# Mobile Laser-Induced Breakdown Spectroscopy for Future Application in Precision Agriculture—A Case Study

**DOI:** 10.3390/s23167178

**Published:** 2023-08-15

**Authors:** Alexander Erler, Daniel Riebe, Toralf Beitz, Hans-Gerd Löhmannsröben, Mathias Leenen, Stefan Pätzold, Markus Ostermann, Michal Wójcik

**Affiliations:** 1Physical Chemistry, University of Potsdam, Karl-Liebknecht-Str. 24-25, 14476 Potsdam, Germany; aerler@uni-potsdam.de (A.E.); loeh@chem.uni-potsdam.de (H.-G.L.); 2Soil Science and Soil Ecology, Institute of Crop Science and Resource Conservation (INRES), University of Bonn, Nussallee 13, 53115 Bonn, Germany; 3Process Analytical Technology, Federal Institute for Materials Research and Testing (BAM), Richard-Willstätter-Straße 11, 12489 Berlin, Germany; markus.ostermann@bam.de; 4Department of Field Theory, Electronic Circuits and Optoelectronics, Faculty of Electronics, Photonics and Microsystems, Wroclaw University of Science and Technology, Wybrzeze Wyspianskiego 27, 50370 Wroclaw, Poland; michal.wojcik@pwr.edu.pl

**Keywords:** LIBS, precision agriculture, soil, multivariate methods, feature selection

## Abstract

In precision agriculture, the estimation of soil parameters via sensors and the creation of nutrient maps are a prerequisite for farmers to take targeted measures such as spatially resolved fertilization. In this work, 68 soil samples uniformly distributed over a field near Bonn are investigated using laser-induced breakdown spectroscopy (LIBS). These investigations include the determination of the total contents of macro- and micronutrients as well as further soil parameters such as soil pH, soil organic matter (SOM) content, and soil texture. The applied LIBS instruments are a handheld and a platform spectrometer, which potentially allows for the single-point measurement and scanning of whole fields, respectively. Their results are compared with a high-resolution lab spectrometer. The prediction of soil parameters was based on multivariate methods. Different feature selection methods and regression methods like PLS, PCR, SVM, Lasso, and Gaussian processes were tested and compared. While good predictions were obtained for Ca, Mg, P, Mn, Cu, and silt content, excellent predictions were obtained for K, Fe, and clay content. The comparison of the three different spectrometers showed that although the lab spectrometer gives the best results, measurements with both field spectrometers also yield good results. This allows for a method transfer to the in-field measurements.

## 1. Introduction

Spatial variations of soil properties, especially of different chemical soil parameters, on agricultural land are relevant for soil fertility and, thus, for crop yields. Chemical soil fertility parameters include the reservoir, i.e., the total contents of the macronutrients such as potassium, magnesium, calcium, nitrogen, and phosphorus; the soil organic matter (SOM) content; and the soil pH value. An important physical soil parameter is the soil texture. The common uniform fertilization of fields can lead to the partial over- or under-dosing of nutrients in parts of a given field. A way to avoid this is the concept of precision agriculture. This concept is based on the measurement of spatial nutrient variability within a field using sensors and circumvents the time-consuming and expensive soil sampling with subsequent laboratory analysis [1]. However, only a few sensor technologies, such as geoelectrical, potentiometric pH, gamma ray, and spectral optical sensors, are currently used [2,3,4].

The ideal method for the determination of nutrient elements in soils is optical emission spectroscopy (OES). However, it requires as a preliminary step the expensive and time-consuming sample preparation via extraction or digestion procedures. A variant of OES that avoids these disadvantages is laser-induced breakdown spectroscopy (LIBS) [5,6,7,8]. LIBS is based on the ablation of material and the formation of microplasma using a laser, typically with nanosecond pulse duration. The plasma excited atoms and atomic ions emit radiation specific to the elemental composition of the sample. The direct (in situ) analysis without sample preparation, the high measurement speed, and the access to the light elements make LIBS potentially a suitable tool for cost-effective and fast on-site analyses in the context of precision agriculture.

While there are currently no publications on large-scale, multi-parameter field mapping via LIBS, the detection of soil nutrients in the laboratory has already been demonstrated in a series of publications [9,10,11]. A good overview of the literature on the application of LIBS in agriculture can be obtained from several review articles recently published [12,13]. The focus of most articles is on the determination of the total contents of metal and non-metal nutrients. Diaz et al. [14] demonstrated that good results with coefficients of determination R^2^ > 0.85 can be obtained via univariate analysis for the determination of Ca, Mg, P, Fe, and Na. Nicolodelli et al. [15] demonstrated that LIBS performance can be improved via the application of double laser pulses instead of single-pulse LIBS. The importance of different machine learning methods, such as artificial neural networks (ANN) [16], partial least squares (PLS), and support vector machine (SVM) regression [17] for the determination of the total contents of various elements was intensively investigated.

In recent years, a shift in focus towards the determination of other soil parameters took place. Special attention was directed to the determination of the organic and inorganic carbon contents in agricultural soils. These investigations started with a univariate analysis of the total carbon content using the two C lines at 193.07 nm and 247.86 nm [18]. Later, multivariate methods were applied [19], which also allowed for the differentiation of inorganic and organic C. The determination of further soil parameters, which are not causally related to the content of a single element, is based on multivariate approaches. These soil parameters include the soil pH [20], the soil texture [21,22], and plant-available nutrient contents [23].

Most work dedicated to LIBS applications on agricultural soils, published in recent years, was based on multivariate methods. The prevalence of these methods is due to the spectral matrix dependence of the LIBS signal. This is influenced by the sample ablation and the plasma excitation, which are both highly complex phenomena. Multivariate analysis of whole spectra intrinsically takes matrix effects into account. The linear multivariate method most often used is partial least squares (PLS) regression. However, PLS regression has limitations. If all spectral channels are used, many channels only contribute noise. A method with the ability to reduce the number of input variables (spectral channels) is the least absolute shrinkage and selection operator (Lasso) regression. Lasso is very similar to least squares regression, except that it includes a penalty term of the l1 norm, which has the property of forcing most coefficients to zero. Multivariate analysis via PLSR and Lasso was already evaluated for LIBS investigations of geological samples [24]. Two non-linear, non-parametric regression methods that were already applied on LIBS spectra are SVM and Gaussian process (GP) regression; both are kernel-based techniques and were already applied on LIBS spectra of soils [9,17]. If a regression model includes all wavelengths of the spectrum, it contains noise and redundant information. This can have a negative influence on the prediction results. Feature selection methods can drastically reduce the number of variables with a minor loss of information. Various feature selection methods were already applied in LIBS data treatment. For this study, two methods were selected, which were successfully applied in LIBS: competitive adaptive reweighted sampling (CARS) [25] and principal component analysis (PCA).

In our earlier article [9], we obtained good results with handheld LIBS for the total contents of Ca, K, and Mg as well as for the soil pH without the necessity of using feature selection. This first study was conducted on an arable field near Berlin with sandy soil. In this present work, a field near Bonn with higher SOM and clay content in a geologically completely different region was investigated. The emphasis was now on (1) the comparison of the regression results of the soil parameters on different soils studied in the previous and this paper, (2) the extension of the investigation to further soil parameters, (3) methodical extensions (SVM, CARS), and (4) the comparison of two field spectrometers with a lab benchtop instrument. While the handheld spectrometer allows for single-point measurements, the second spectrometer is intended for future applications on a mobile sensor platform. In the first part, the total contents of N, P, K, Ca, Mg, Cu, Zn, Fe, and Mn, as well as the soil pH, the SOM content, and the soil texture (silt and clay contents), were determined. In addition to standard normal variate (SNV) normalization after background correction, the methodical extensions include the usage of the non-linear, non-parametric SVM regression and investigations of two methods of variable selection (CARS and PCA). Furthermore, the focus was on the interpretability of our prediction models, especially on the identification of the lines (elements), which contribute to the determination of soil parameters such as soil pH, SOM content, and soil texture.

## 2. Experimental Part and Data Analysis

### 2.1. Materials

The study field at Bölingen near Bonn (Germany) has a size of 2.8 ha and revealed large soil heterogeneity. Soil samples were taken from the cultivated topsoil (Ap horizon, 0–0.3 m depth) along a regular raster of 21 m grid size (in total 68 samples). The soils at Bölingen developed in Pleistocene periglacial slope deposits, generally complex parent materials that consisted in case of the Bölingen field of weathered Lower Devonian sandstones, siltstones, claystones, and Weichselian loess. The loess proportion in the topsoil differed largely, leading to variable soil texture, stone content, and mineralogy. Moreover, the weathering degree of the Lower Devonian material in the topsoil was very variable due to Pleistocene mixing. The complex substrate genesis made the sample set very heterogeneous in terms of element contents, pH value, and soil texture. The large clay contents in some parts of the field were correlated to large SOM contents. The results for the Bölingen sample set will be compared to two fields from a preliminary study [9,10]. Note that the two sites from the former study are located in geologically very different landscapes and soils developed from largely different, Weichselian parent materials: glaciofluvial sand at Wilmersdorf and ground moraine marl at Booßen.

### 2.2. Reference Analysis

The total element contents were determined using wavelength dispersive X-ray fluorescence spectroscopy (WDXRF) to create reference values and for validation purposes. The air-dried and sieved (2 mm) Bölingen samples were characterized as loose powder using the WDXRF spectrometer MagixPro (Panalytical, Malvern, UK). The soil samples were placed in 40 mm wide X-ray sample cups covered with a 6 μm thick polypropylene foil. Helium was used as atmosphere for the measurements. The calibration of the WDXRF was conducted using the instrumental software SuperQ 5.1B. The measurement application and the calibration model were created using 16 certified reference soil materials (CRM) from different certification institutions:

GBW07402 and GBW07405 from National Research Centre for Certified Reference Materials (Beijing, China); NCS DC73023, NCS DC73030, NCS DC85109, and NCS DC87104 from National Analysis Centre for Iron and Steel (Beijing, China); TILL1, TILL2, and TILL3 from Canadian Centre for Mineral and Energy Technology—CANMET (Ottawa, ON, Canada); BAM-U110 from Federal Institute for Materials Research and Testing—BAM; Soil-5 from International Atomic Energy Agency—IAEA (Vienna, Austria); VS2498-83 from ICRM-Centre (Moscow, Russia); NIST1646a, NIST2704, NIST2709, and NIST2710 from National Institute of Standards and Technology—NIST (Gaithersburg, MD, USA). The certified elemental concentrations of the reference materials are published on the websites of the indicated institutions or can be supplied on demand.

The further soil properties under study were determined using standard procedures. Soil texture was characterized using the sieve and pipette method, the pH values were potentiometrically measured in 0.01 M CaCl_2_, and SOM was determined using elemental analysis via the total C content, where total C equaled organic C in the carbonate-free samples.

### 2.3. LIBS Measurements

**Sample preparation.** The soil samples were air dried, pestled, and sieved (2 mm mesh). Soil pellets were produced by taking 3 g of dried soil sample and mixing it with 90 µL of water to establish a standardized moisture. Then, the soil samples were homogenized using a ball mill (MM 400, Retsch, Hahn, Germany) and pressed into pellets at 80 kN without applying binding agents (TP 40, Herzog Maschinenfabrik, Osnabrück, Germany).

**LIBS measurement procedure.** The pellets were measured using an echelle spectrometer (Aryelle Butterfly, LTB, Berlin, Germany). This lab benchtop (Lab) spectrometer has two separate measurement ranges: the UV and VIS spectral regimes. The VIS range was used in this work and covers the range from 278 to 769 nm with a resolution between 25 and 60 pm. An iCCD camera (iStar, Andor Technology, Belfast, UK) was used as detector. Five laser shots were accumulated. Apart from the lab benchtop instrument a spectrometer to be potentially used on a sensor platform and a handheld (HH) spectrometer were applied. The HH instrument has a detection range of 190–950 nm and is equipped with a laser of 1064 nm at ca. 6 mJ. Here, the samples are measured in an 8 × 8 grid (64 shots over 1 mm^2^ surface area) at a frequency of 10 Hz. For the sensor platform, a multichannel CMOS spectrometer (AvaSpec Mini 2048CL, Avantes B.V., Apeldoorn, The Netherlands), that is further referred to as platform (PF) spectrometer as well as a handheld Spectrometer (HH; Z-300, SciAps, Woburn, MA, USA) for single-point measurements, were used. The PF spectrometer consists of four modules with wavelength ranges of 210–328 nm, 325–476 nm, 473–700 nm, and 695–890 nm, respectively. The laser (Bernoulli LIBS 150-25, Litron Lasers, Warwickshire, UK) used for the measurements in the Lab and the PF spectrometer emits radiation at a wavelength of 1064 nm, with a repetition rate of 25 Hz, a pulse duration of 10 ns and pulse energy attenuated to 40 mJ. The pellets were rotated and linearly translated during measurements on a sample holder forming a spiral-like trace of ablation events. Thus, each spectrum is measured on a new point of the pellet. A total of 200 spectra were recorded per sample in the VIS range. Emissions were collected via a CC52 Collector/Collimator (Andor Technology, Belfast, UK), coupled into an optical fiber, and guided to the Lab or PF spectrometer. Optimization of LIBS spectra acquired using the Lab spectrometer led to the following measurement parameters: a detection delay of 2 µs, a measurement window of 10 µs, as well as a constant amplification factor of the iCCD camera of 2000.

### 2.4. Data Analysis

**Data preparation.** The following procedure of spectra preparation was applied. First, the remaining background, which is still formed by the continuous radiation after a delay of 2 µs, was removed. This procedure is based on the top-hat filter and was implemented in Matlab (imtophat function, square as structure element). Then, all spectra were normalized via standard normal variate (SNV) normalization. Finally, all spectra within one sample (pellet) were averaged. In order to identify any existing outliers in the data set, the robust principal components analysis (ROBPCA) method of Hubert et al. [26] was applied.

**Feature selection.** As mentioned in the introduction, high-resolution Lab, PF, and HH spectra consist of 37,633; 7744; and 7810 data points and contain redundant information (e.g., noise). The methods tested for feature selection are CARS and PCA. Additionally, a very simple method, referred to as threshold method in this work, cuts parts of the full spectrum below a selected threshold. The remaining part of the spectrum above the threshold is then used for regression.

**Competitive adaptive reweighted sampling (CARS).** The CARS algorithm is computationally intensive and consists of 4 steps that are iteratively carried out [25]. In the first step, the samples for a PLS model are randomly selected. PLS is carried out and a normalized weight for evaluating the importance of each wavelength is calculated. In the second step, the number of wavelengths is reduced sequentially using an exponentially decreasing function. In the third step, adaptive reweighted sampling is employed to eliminate the wavelengths in a competitive way. This step considers the calculated weights in step 1. In the fourth step, the RMSECV is calculated. After N sampling runs, N subsets of wavelengths and the corresponding RMSECV are obtained. In the final step, the subset with the lowest RMSECV is selected. The RMSECV-Monte Carlo sampling number curve often has several local minima, in addition to the global minimum, which are representing different-sized subsets of wavelengths. CARS often generates very compact feature sets compared to other feature selection methods.

**Principal component analysis**. The PCA algorithm will find a linear combination of the given wavelengths (features) with the highest variance and set it as the first PCA component. The subsequent components are found by looking for the highest variance linear combination in the subspace orthogonal to their previous components. As a result, uncorrelated linear projections of the original data are obtained. In PCA, to overcome the under- and over-fitting of the regression model with the resulting components, the largest variance criteria together with the heuristic approach called elbow method are applied. That way, a point in the explained variance plot is found, after which only diminishing returns are obtained. Those were neglected for further analysis since the corresponding components do not carry additional, meaningful information.

**Multivariate methods.** Four multivariate methods, namely PLS, Lasso, GP, and SVM regression, were used for obtaining calibration models. **PLS** regression, which has often been described in detail, is widely applied in the LIBS community and can be regarded as a reference method. **Lasso** is a linear regression method that constrains the coefficient estimates and shrinks those that do not significantly contribute to the correlation to zero. This enables a robust linear regression and a simplified interpretation of the coefficients. In this work, the number of coefficients was always reduced to the number necessary for an error one standard deviation above the minimum (one standard error, 1SE). In addition to these two linear approaches, two non-linear regression methods were also used. Popular non-linear regression methods are SVM and GP regression. In contrast to PLS and Lasso regression, **GP** regression is a nonparametric and kernel-based Bayesian approach. It is a local regression approach, which uses a kernel for weighting neighboring observations in the estimation. Compared to other kernel methods, such as splines and support vector machines, GP regression is slower but yields properly tuned probabilistic outputs and is sometimes more robust and flexible. An additional benefit of GP regression is its good suitability for small datasets. **SVM** regression is also considered a nonparametric technique because it relies on kernel functions similar to GP regression. SVM regression tries to find an appropriate line (or hyperplane in higher dimensions) to fit the data. The algorithm uses margins that are controlled by a hyperparameter ε. The aim is to fit as many instances as possible inside the margins while limiting margin violations. For any value that falls outside of the margins, its deviation from the margin can be described by slack variables. These are part of the objective function that will be minimized. Nonlinear regression tasks will be solved using a kernelized SVM model. One of the main advantages of SVM regression is that its computational complexity does not depend on the dimensionality of the input space. Additionally, it has an excellent generalization capability with high prediction accuracy. All methods were implemented in Matlab (Version 2020a, MathWorks, Natick, MA, USA). PLS regression was based on plsregress, Lasso regression on lasso, GP regression on fitrgp, and SVM regression on fitrsvm. All functions are included in Matlab’s Statistics and Machine Learning Toolbox.

**Validation.** Different validation procedures were tested. All multivariate methods were 10-fold cross-validated. The determination of the RMSECV and the coefficients of determination is based on this 10-fold cross-validation. Additionally, the data were randomly split into a 70% training (N_train_ = 48) and a 30% test (unseen data, N_test_ = 20) data set using the cvpartition function in Matlab. The test data were not used for the feature selection methods and training of the multivariate model. Furthermore, stratified sampling was used for validation that reduces the variance by constraining a proportion of the samples to specific subsets of the sample space (percentiles of the specific soil parameter).

**Interpretability.** Especially for the determination of soil pH, soil texture, and SOM content, the influence of the individual lines (elements) and the identification of the most important lines is interesting. Two metrics were used to identify these lines: the PLS weights and the variable importance for the projection (VIP) score. The PLS weights are part of the PLS result, and the weights of the first principal components were considered. The VIP score attributes each wavelength (feature) a measure of its contribution to the PLS regression. The VIP value [27] is defined in Equation (1) as follows:(1)$$VIP_{j}=\sqrt{\frac{\sum_{f=1}^{F}w_{jf}^{2}\cdot SSY_{f}\cdot J}{SSY_{total}\cdot F}}$$

With *w_jf_* as the weight value for the wavelength *j* and the component *f*, *SSY_f_* as the sum of squares of explained variance for the *f*th component, *J* as the number of wavelengths, *SSY_total_* as the total sum of squares of explained variance, and *F* as the total number of components. The VIP values reflect how important the information of each wavelength is for the PLS regression. There is an arbitrary threshold, which is often selected as 1.

## 3. Results

### 3.1. Determination of Soil Parameters

In the following section, the determination of the total nutrient contents, the soil pH, SOM content, and the soil texture (silt and clay content) is discussed. To achieve a first overview of the potential of LIBS for the determination of different soil parameters, only the results obtained via the echelle (Lab) spectrometer and the best multivariate method are reported. Systematic investigations of different spectrometer types and different methods of data analysis are discussed in later sections.

**Total contents of metal macronutrients.** The metal macronutrients K, Ca, and Mg belong to the alkali and alkaline earth metals, which can be sensitively detected in the laser-induced plasma. The three elements have their strongest lines in the wavelength range of the Lab spectrometer between 278.4 and 768.9 nm (Ca: 393.36 nm, K: 766.5 nm, Mg: 279.56 nm). The concentration range of the three nutrients is between 0.44 and 1.96 w%. The median of the total nutrient contents increases from 0.59 w% for Ca over 0.65 w% for Mg to 1.71 w% for K. Ca has the greatest variation in the total contents at 86% compared to Mg at 63% and K at 60% (all range 90%). Multivariate regression of the total high-resolution spectra yields good results. The coefficients of determination for cross-validation are 0.80 (for Mg), 0.86 (for Ca), and 0.92 (for K), and for the validation between 0.79 (for Mg) and 0.92 (for K). The order of the coefficients of determination correlates well with, on the one hand, the median and, on the other hand, the variation of the concentration range of the nutrients. K has the largest median, and Ca has the largest concentration range with a comparable median to Mg. LIBS, in combination with multivariate regression, should allow for a good to excellent quantitative determination of the metal macronutrients (Figure 1).

**Total contents of non-metal macronutrients.** Nitrogen and phosphorous are essential, non-metal plant macronutrients, and their content and dynamics in soil depend largely on biological processes. The two elements are difficult to detect within the wavelength range of the echelle spectrometer. Although they have a few lines in this range, their line strengths are weak. The most intense lines are outside of the wavelength range of the spectrometer. Both elements can be detected preferentially in the vacuum ultraviolet (VUV) range. Alternatively, phosphorus can be detected at 213.62 nm. The significantly lower concentration compared to Mg, Ca and K makes their detection even more difficult. The median of N content, 0.13 w%, is slightly larger than the median of P content, 0.09 w%. In contrast, the variation of the P contents with 106% is larger than that of N with 48%. Accordingly, the coefficients of determination for PLS regression of the entire spectra are lower in comparison to the metal nutrients. These vary from R^2^ = 0.71 for N (smaller variation) to R^2^ = 0.82 for P (for validation).

**Total contents of metal micronutrients.** From an agricultural perspective, Zn, Mn, Cu, and Fe are micronutrients, but their total contents in soil differ strongly. While the median of the Fe content is high (3.51 w%), the other three elements are considered trace elements and have, thus, much lower medians. But their medians also vary strongly. Mn has the highest median (0.1 w%). The medians of Zn (0.01 w%) and Cu (0.003 w%) are much lower. For the three trace elements, the variation of the total contents increases with decreasing median: Mn (102%), Zn (167%), and Cu (200%). Compared to the metal macronutrients, these four elements have their most intense lines outside of the wavelength range of the spectrometer: 200–215 nm for Zn, 257–261 nm for Mn, 198–225 nm for Cu, and 233–275 nm for Fe. The lines inside the wavelength range of the spectrometer are much weaker for Zn and Fe. In contrast, Cu and Mn have sufficiently strong lines. Despite the significantly lower concentrations compared to the macronutrients and the fact that the most intense lines are outside of the wavelength range of the spectrometer, the coefficients of determination are similar. They vary between R^2^ = 0.85 for Mn, R^2^ = 0.76 for Zn, R^2^ = 0.87 for Cu, and R^2^ = 0.92 for Fe (validation).

**Additional soil parameters.** For agricultural purposes, further soil properties are relevant, such as soil pH, SOM content, and soil texture (silt and clay content). Though, their determination via LIBS is challenging because they cannot be measured directly. They are based on correlations to certain elements, which are often unknown and difficult to determine. For example, in the case of pH that may be the Ca content, as demonstrated earlier [9,20]. The challenge in measuring SOM content is generally due to the presence of both organic and inorganic carbon in the soil, and the fact that the carbon lines are outside the measurement range of the echelle spectrometer. In Bölingen, the determination of organic carbon is somewhat easier since no inorganic carbon is present here. Soil texture is characterized in this work by determining the clay and silt fractions (Figure 2). The characterization of the clay fraction, for example, is based on the correlation to the elements Al, Fe, and K via LIBS.

The coefficients of determination vary strongly for the different soil parameters (Table 1). Multivariate regression of the total spectra demonstrates that a correlation exists for the soil pH (R^2^ = 0.57). This rather weak result allows only to distinguish between high and low values. In a preliminary study, much better results (R^2^ > 0.9) were obtained for the soil pH; yet, the pH variation at the Wilmersdorf field [9] is much larger than at Bölingen, where the pH variation is 1.15 pH units (range 90%). This can explain the poorer regression result. In the case of the determination of the SOM content (R^2^ = 0.71), approximate quantitative predictions are possible. The regression results are slightly better than for the Wilmersdorf measurements (R^2^ = 0.58). This can be explained by the higher SOM content in Bölingen (median: 2.4 vs. 1.8 w%) and the larger variation in clay content because, in general, SOM tends to show a correlation to clay content. In contrast, LIBS is much better suited for the characterization of soil texture. PLS of the total spectra yields good coefficients of determination for the clay (R^2^ = 0.91) and the silt content (R^2^ = 0.88). This result corresponds to the inorganic character of clay and silt, which are, to a large extent, built up by elements within the LIBS spectra (i.e., Ca, Mg, K, Fe, Mn).

### 3.2. Matrix Influence

**Principal component analysis (PCA).** To obtain an impression of the spectroscopic homogeneity of the soil of the field in Bölingen, the samples from there were compared with samples from two other fields via PCA, which originate from geographically distant areas with different geopedological settings. The red dots in the score plot (Figure 3) of the first two principal components belong to the field in Bölingen. Their area is very small. In contrast, the areas of the fields near Wilmersdorf (green dots) and Booßen (blue dots) are much larger. This illustrates that the field in Bölingen consists of a spectroscopically relatively homogeneous matrix.

In order to compare the homogeneity of the fields over multiple PCs at once a multidimensional Euclidean distance was applied. The single distance calculation between two points is given by Equation (2) as follows:(2)$$D_{ist}=\sqrt{\sum_{1}^{N}{\left(x_{n}-y_{n}\right)}^{2}}$$
where *N* is the number of dimensions set by the number of PCs used. Then, the point-to-point distances calculated between all the possible combinations of the points in the given field are averaged over the corresponding dimensions. In Figure 3, it can be observed that in the Bölingen field, those averaged distances increase much slower with the number of PCs used than in the other two fields. Thus, it is assumed that the spectra taken from this field represent the most homogeneous set of the three.

**Wavelength influence on regression.** It is interesting to know which wavelengths, in particular, contribute to the regression model. In PLS regression, this can be achieved by calculating the PLS weights or using the variable importance in projection (VIP) scores. In the case of determining the total element contents, the question is which lines, in particular, contribute to the regression. This can help to estimate to what extent the regression is sensitive, e.g., line superpositions by other elements. In the case of the determination of soil parameters such as soil pH, SOM content, or clay content, which can only be determined indirectly, the identification of elements that are important for the determination of these soil parameters is of outstanding interest. In addition, indications of soil chemical or soil physical relationships can potentially be obtained. A further aspect is a deeper characterization of the matrix effect. This includes the identification of matrix elements that directly affect the regression model.

Calcium was chosen as an example for the detection of the total content of an element. In the wavelength-dependent representation of VIP scores (not shown) and PLS weights (Figure 4a), eight Ca lines were found that contribute to the regression model. The VIP score also shows the contributions of further elements, such as K, Mg, Cu, Fe, Na, Si, Al, and Ti, which are part of the soil matrix. For examining the PLS weights, only the first two PCs (PC I and PC II) were considered because they explain a large part of the total variance. The PLS weights of each element can be classified into three subgroups. While Fe, Na, and Ti have PLS weights with positive and negative signs, Al, Si, and K possess negative PLS weights, and Mg and Ca possess only positive PLS weights. Therefore, Ca can be determined directly by correlating its lines and the lines of Mg in the PLS regression. The matrix influence is manifested, in particular, by the PLS weights of the elements Al, Si, K, Fe, Na, and Ti.

The soil pH can be indirectly determined in LIBS by correlations to specific elements. These are alkaline earth metals (Mg, Ca) with lines appearing in both VIP scores and PLS weights (Figure 5a, Figure 4b). These have exclusively positive PLS weights in the first two components. In addition, other elements of the matrix, such as Si, Al, Fe, and Ti, are observed with weights predominantly having a negative sign. From a pedological perspective, the correlation between total element content, irrespective of the element, and pH is not due to a direct causality. In most soils, the total element contents are by magnitudes larger than the dissolved and exchangeable concentrations, which determine the pH value.

The determination of the clay content as part of the soil texture is based on the determination of the elements contained in the clay. These elements should have positive weights (Figure 4c). The main contribution in PC II is from Mg, Ca, Si, and Al, which have positive weights. In PC I, especially Mg, Ca, Si, and Fe show positive weights. These elements are constituents of different clay minerals (Ca and Mg adsorbed).

Similar to soil pH, SOM cannot be directly determined via LIBS because the two most intense C-lines do not lie within the measuring range of the Echelle spectrometer. In addition, a distinction must be made generally between organically and inorganically bound carbon (not in Bölingen, see above). Negative PLS weighs were found for elements that are part of the inorganic matrix (soil minerals), whereas positive PLS weights can be assigned to alkaline earth metals (Ca, Ba, and Sr), maybe via the adsorption of organic functional groups by Me^2+^ (Figure 4d).

### 3.3. Different Aspects of Measurement and Data Analysis

**Comparison of different spectrometers.** Precision agriculture requires on-field measurements. These can include single-point measurements using a robust and less expensive handheld spectrometer, but also much faster measurements on a sensor platform allowing the spatially resolved mapping of soil parameters on whole fields. To obtain an impression of the performance of potential LIB spectrometers for these application profiles, two instruments (HH and PF spectrometer) were characterized in comparison to a high-resolution Lab instrument. The main differences between the spectrometers with respect to LIBS measurements are the resolution and sensitivity of the spectrometers and the pulse energy of the excitation laser. For example, the resolution of the lab benchtop spectrometer is 20–30 pm, while the other two instruments have resolutions of 100–250 pm. Pulse energies vary from 6 mJ for the HH spectrometer to 40 mJ in the case of the Lab and PF spectrometers. The instruments also differ in other parameters, such as costs, size, weight, robustness, and measurement speed. We take as a metric of comparison the figures of merit of the regression models (in this case the coefficients of determination) for the different soil parameters. The results are summarized in Table 2.

Table 2 shows the results of the comparison of different spectrometers (Lab, PF, HH) and plasma excitation conditions (laser pulse energies: 6, 10, 25, and 40 mJ), as expressed by the coefficients of the determination for the different soil parameters. The best figures of merit are, in most cases, obtained for the Lab spectrometer with the highest excitation energy. Then, they decrease from the platform to the handheld spectrometer. The reduction in the pulse energy from 40 to 25 mJ in the case of the PF spectrometer did not lead to significant degradation of the coefficients of determination. The strongest effect was observed for the trace nutrients. The further reduction in pulse energy from 40 to 10 mJ in the case of the Lab instrument decreases the R^2^ on the level of the handheld spectrometer. The strongest degradation of R^2^ is observed for the soil parameter pH. The strength of the Lab spectrometer lies in the better detection of the non-metallic nutrients (N, P) and the soil parameters SOM, pH, silt, and clay. Since the main difference between the HH and PF spectrometers is the resolution, the separation of the different lines seems to be of great importance.

In conclusion, the best results were obtained using the Lab instrument with high pulse energy. This is based on the combination of high resolution by the echelle grating, high excitation pulse energy, and sensitive detection via the ICCD camera. The PF spectrometer on the basis of four Czerny–Turner spectrometer modules is more robust and allows for, in combination with a faster laser, nearly 100 times faster measurements (100 Hz) than the lab instrument. These are important prerequisites for nutrient mapping of large fields. The handheld spectrometer, which has advantages in size, weight, and price, demonstrates its potential for point measurements on the field. The results indicate that a reliable recording of soil properties using mobile LIBS instruments is possible and bears great potential for soil mapping.

**Comparison of regression methods.** In order to identify and eliminate any existing outliers in the data set, the robust principal components analysis (ROBPCA) method of Hubert et al. [26] was applied. As a result, an outlier map (Figure 6) was obtained, which allows for the detection of any existing outliers. Here, the orthogonal distance is plotted against the score distance. The two upper quadrants contain the potential outliers, the so-called bad leverage points, and the orthogonal outliers. In particular, the orthogonal outliers should be removed.

In the case of the Bölingen spectra, three samples (13, 31, 52) can potentially be removed as outliers. The effect of removing individual outliers on the result of the PLS regression of selected soil parameters was investigated. The orthogonal distance is not very large. Accordingly, the effect of the distance of the outliers on the PLS results is also small. For some soil parameters, a slight improvement and, for others, a slight deterioration of the results can be observed. Therefore, we have decided not to remove the potential outliers.

In addition to PCR and PLS regression as linear, parametric multivariate regression methods, SVM and GP regression as nonlinear, nonparametric regression methods were also investigated. The regression results are displayed in Table 3. Coefficients of determination of cross-validation and validation of the test data set are given as metrics of comparison.

Studying a full spectrum with its 37,633 data points is the most challenging task for the regression methods. The calculation of the average coefficients from the determination of the four methods shows that similar results were obtained via PLS (R^2^ (CV) = 0.79), PCR (R^2^ (CV) = 0.76), SVM (R^2^ (CV) = 0.74), and GP (R^2^ (CV) = 0.79). The correlation between full spectra and concentrations seems to be linear, with some exceptions. This assumption is based on the observation that PLS, as a linear regression method, as well as GP, as a non-linear regression method, yield similar results. Exceptions were Mg, P, and N, where GP regression achieves better results than the linear regression methods PLS and PCR. This is also reflected in the better coefficients of determination of all macronutrients. Here, GP regression yields slightly better results on average than PLS, PCR, and SVM regression. This suggests that GP regression is slightly better at dealing with complex data, which contains noise and non-informative data points.

**Feature selection.** The combination of only 68 labeled soil samples and 37,633 features (Lab spectrometer) is a challenging regression task. Feature selection methods can overcome this problem by removing non-informative wavelengths, which only contribute to noise. This also improves the interpretability of the regression models. In this work, different feature selection methods based on different algorithms are applied, such as PCA, CARS, and Lasso. In addition to the methods mentioned above, a very simple method, referred to as the threshold method in this work, cuts parts of the full spectrum below a selected threshold. The remaining part of the spectrum above the threshold is then used for PLS regression. A systematic increase in the threshold reduces the size of the spectra and, of course, reduces the information available. The results of the PLS regression of the determination of Ca contents after feature reduction using the threshold method are shown in Figure 7 for different sizes of spectra. As the size of the spectra decreases, a maximum of the coefficients of determination is first reached, which transitions to a plateau and only declines sharply when the number of features is very small.

The results of the feature selection methods (PCA, Cars, Lasso) are shown in Table 4. While CARS reduces the number of features by more than a factor of 18, from 37,633 data points to an average of 2042, PCA uses 17 and 242 features (principal components), which are linear combinations of all features (wavelengths), respectively. This is associated with very slight improvements in the mean coefficients of determination obtained via PLS regression from R^2^ = 0.76 of full spectra to an average of R^2^ = 0.77 of both PCA-reduced spectra and R^2^ = 0.80 of CARS-reduced spectra. A closer look shows that CARS, in particular, improves the determination of Mg, P, and the soil pH. In most cases, similar results are obtained via PCA and CARS with greatly reduced data size. Furthermore, it is noteworthy that both PCA-reduced spectra, regardless of the different sizes, yield almost equal results.

The Lasso method combines feature selection and regression. In comparison to PLS regression, the application of Lasso regression on the full spectrum with 37,633 data points leads to worse coefficients. But these are based on significantly stronger feature reduction in the full spectrum with an average of 15 features in comparison to CARS and PCA. The main advantage of Lasso is the better interpretability of the prediction model. The strong reduction in the number of features enables the identification of the most important lines (elements) contributing to the prediction model. While, in PCA, the PCs are linear combinations of all wavelengths, in Lasso, most wavelengths are set to zero and do not contribute to the model. This strong reduction is, of course, the cause of the somewhat worse results. It can also be expected that due to the stronger restrictions, overfitting is better avoided.

## 4. Conclusions

The total contents of important macro- and micronutrients, as well as soil pH, SOM, silt, and clay content, can be determined with good results by investigating soil pellets with laser-induced breakdown spectroscopy. This was demonstrated in a case study on a heterogeneous arable field near Bölingen and is the prerequisite for the transfer of LIBS analytics from the laboratory to the field. Applying LIBS directly in the field offers great opportunities for precision agriculture. To address this in more detail, two spectrometers that can be used in the field, a handheld spectrometer for point measurements and a spectrometer mounted to a mobile platform for mapping larger areas, were compared to a high-resolution laboratory spectrometer. Although the laboratory spectrometer yielded the best results, the large potential of the two field spectrometers was demonstrated. Since matrix effects strongly influence LIBS results, machine-learning methods must be applied. It turned out that the four methods, PCR, PLS, SVM, and GP regression, lead to similar results. The applications of PLS weights, PLS VIP scores, and Lasso coefficients allow us to identify the most important wavelengths and interpret the regression models. PCA enables to compare the heterogeneity and similarity of the field to other fields. Feature selection can slightly improve the regression results.

## Figures and Tables

**Figure 1 sensors-23-07178-f001:**
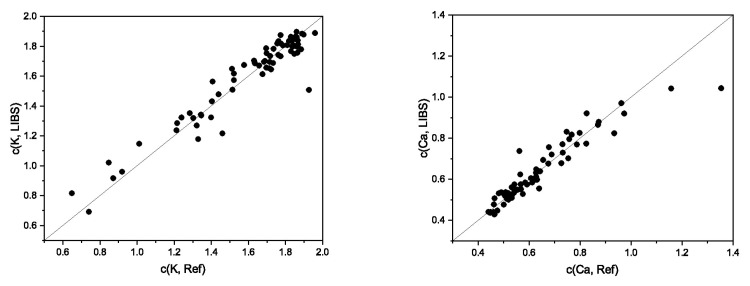
Correlation of predicted contents (LIBS) and total contents obtained via reference analysis (WDXRF) of (**left**) K and (**right**) Ca; GP regression of full echelle spectra, values in w%.

**Figure 2 sensors-23-07178-f002:**
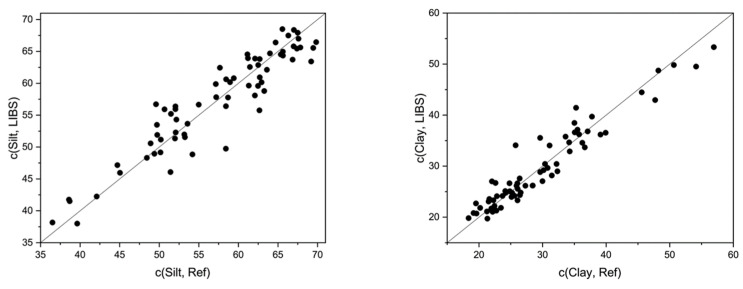
Correlation of measured (reference analysis) and predicted (LIBS) silt (**left**) and clay (**right**) contents, GPR of full echelle spectra, values in %.

**Figure 3 sensors-23-07178-f003:**
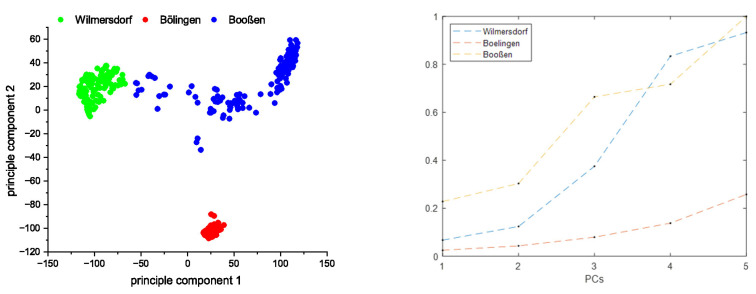
(**left**) PCA of three different agricultural fields which are situated in different geological landscapes of Germany and represent different soil parent materials; (**right**) averaged point-to-point distance for the corresponding fields calculated in N-dimensional PC space. All distances were normalized to the maximum value of the set.

**Figure 4 sensors-23-07178-f004:**
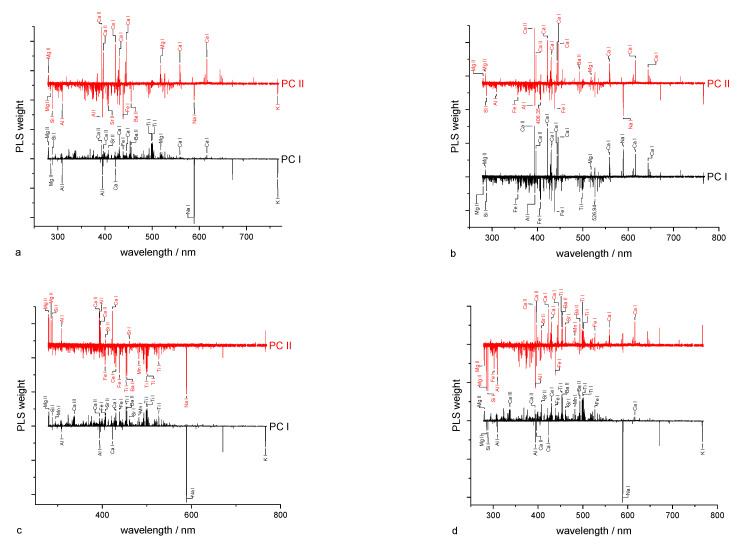
Weights of PLS regression of (**a**) Ca content, (**b**) soil pH, (**c**) clay content, and (**d**) humus content.

**Figure 5 sensors-23-07178-f005:**
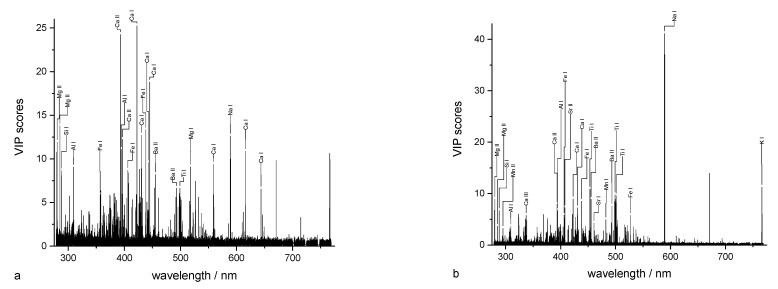
VIP scores of PLS regression for (**a**) soil pH and (**b**) clay content.

**Figure 6 sensors-23-07178-f006:**
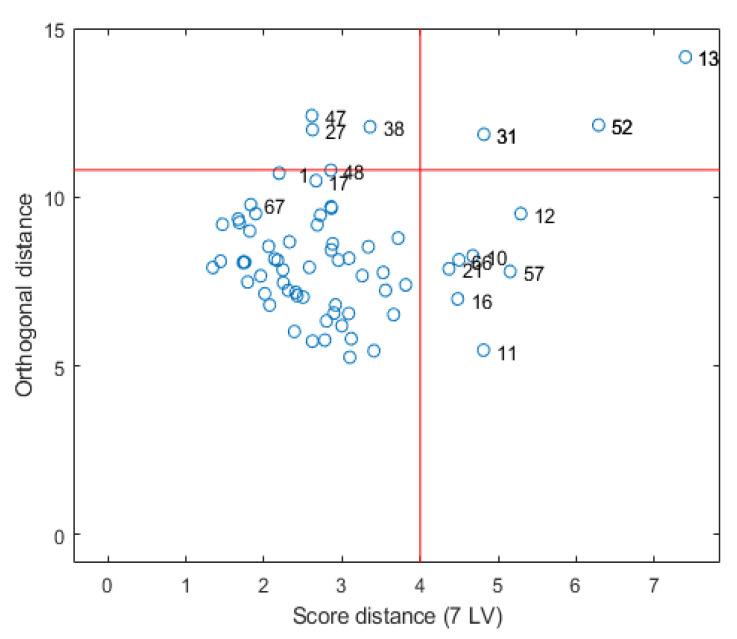
Outlier map obtained via the robust principal components analysis (ROBPCA) method [26].

**Figure 7 sensors-23-07178-f007:**
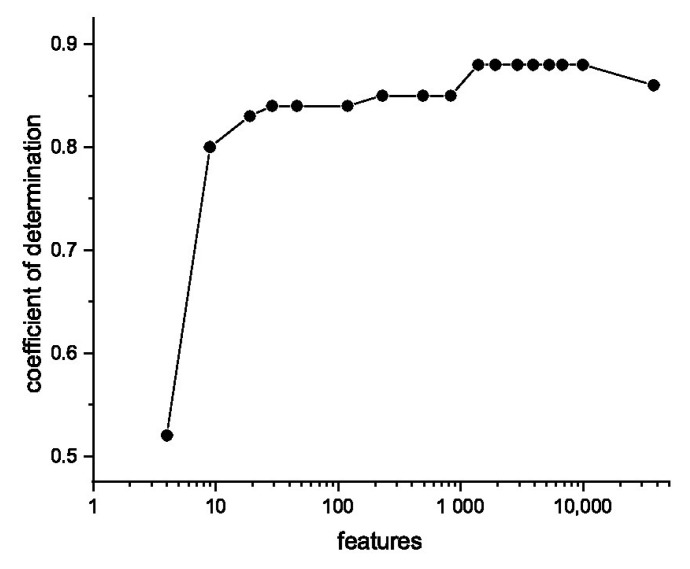
Effect of feature reduction using the threshold method on the coefficient of determination for the Ca prediction via GP regression.

**Table 1 sensors-23-07178-t001:** Soil parameters and descriptive statistics (median, interquartile range (IQR), range between 95th and 5th percentile, which includes 90% of all sample points, all in w%, except pH) as well as an overview of coefficients of determination (R^2^) and RMSECV for various soil parameters, which are the result of the regression of predicted contents (LIBS, lab spectrometer, 40 mJ) and contents obtained via reference analysis (WDXRF and lab soil analysis). Results of the best regression method. (SP: soil parameters, CV: cross-validation, Val: validation (70/30)).

SP	Method	Median	IQR	Range (90%)	R^2^ (CV)	R^2^ (Val)	RMSECV	RMSEV
Ca	GP	0.59	0.21	0.50	0.86	0.86	0.0645	0.0775
Mg	GP	0.65	0.16	0.41	0.80	0.79	0.0561	0.0602
K	GP	1.71	0.43	1.02	0.92	0.92	0.0901	0.0655
P	GP	0.09	0.04	0.10	0.77	0.82	0.0174	0.0184
N	GP	0.13	0.02	0.06	0.62	0.71	0.0118	0.011
Fe	GP	3.51	1.71	4.52	0.93	0.92	0.3489	0.431
Mn	PLS	0.10	0.04	0.10	0.79	0.77	0.0189	0.0203
Zn	PLS	0.01	0.01	0.02	0.87	0.84	0.0018	0.002
Cu	PLS	0.003	0.002	0.006	0.87	0.85	6.62 × 10^−4^	7.11 × 10^−4^
SOM	GP	2.39	0.57	1.43	0.67	0.71	0.2583	0.227
pH	GP	6.07	0.27	1.15	0.54	0.57	0.2203	0.2343
Silt	GP	58.57	12.34	28.43	0.83	0.89	3.4688	3.1757
Clay	GP	26.51	11.86	28.84	0.91	0.91	2.6018	3.0933

**Table 2 sensors-23-07178-t002:** Comparison of different spectrometers and plasma excitation conditions by coefficients of determination (R^2^ (CV)), which are the result of the regression between predicted contents (LIBS) and contents obtained via reference analysis (WDXRF and lab soil analysis). HH: handheld spectrometer, Lab: lab benchtop spectrometer, PF: platform spectrometer, laser excitation energies in mJ, ML method: GP regression.

Soil Parameter	HH	Lab/10 mJ	Lab/40 mJ	PF/25 mJ	PF/40 mJ
	R^2^ (CV)				
Ca	0.85	0.74	0.86	0.91	0.95
K	0.96	0.92	0.92	0.86	0.95
Mg	0.88	0.60	0.79	0.89	0.84
P	0.69	0.63	0.82	0.75	0.62
N	0.59	0.64	0.71	0.30	0.59
Mn	0.69	0.68	0.77	0.40	0.94
Fe	0.94	0.87	0.92	0.87	0.88
SOM	0.55	0.63	0.71	0.65	0.35
soil pH	−0.06	0.10	0.57	0.31	0.55
silt	0.86	0.84	0.89	0.85	0.83
clay	0.83	0.86	0.91	0.93	0.94

**Table 3 sensors-23-07178-t003:** Comparison of the regression methods PLS, PCR, SVM, and GP for different soil parameters (Lab spectrometer).

	PLS		PCR		SVM		GP	
Soil Parameter	R^2^ (CV)	R^2^ (Val)	R^2^ (CV)	R^2^ (Val)	R^2^ (CV)	R^2^ (Val)	R^2^ (CV)	R^2^ (Val)
Ca	0.86	0.84	0.82	0.82	0.82	0.82	0.86	0.86
K	0.92	0.91	0.92	0.93	0.91	0.93	0.92	0.92
Mg	0.77	0.74	0.74	0.76	0.74	0.76	0.8	0.79
P	0.72	0.69	0.71	0.71	0.73	0.73	0.77	0.82
N	0.49	0.48	0.55	0.68	0.64	0.7	0.67	0.75
Cu	0.87	0.87	0.87	0.86	0.65	0.56	0.81	0.76
Mn	0.81	0.85	0.72	0.78	0.72	0.85	0.72	0.76
Zn	0.85	0.76	0.82	0.69	0.58	0.59	0.82	0.66
Fe	0.93	0.98	0.93	0.92	0.93	0.92	0.93	0.92
Humus	0.64	0.62	0.63	0.69	0.62	0.74	0.67	0.71
pH	0.54	0.39	0.48	0.52	0.52	0.49	0.54	0.57
Silt	0.88	0.86	0.83	0.88	0.84	0.89	0.83	0.89
Clay	0.93	0.9	0.92	0.89	0.92	0.9	0.91	0.91
Mean (K, Ca, Mg)	0.85	0.83	0.83	0.84	0.82	0.84	0.86	0.86
Mean (MN)	0.75	0.73	0.75	0.78	0.77	0.79	0.80	0.83
Mean (TN)	0.87	0.87	0.84	0.81	0.72	0.73	0.82	0.78
Mean (SP)	0.75	0.69	0.72	0.75	0.73	0.76	0.74	0.77
Mean	0.79	0.76	0.76	0.78	0.74	0.76	0.79	0.79

**Table 4 sensors-23-07178-t004:** Comparison of different feature reduction methods for determination of soil parameters (SP), PLS as regression method and reference, Lab spectrometer.

SP	Method	Features	R^2^ (CV)	R^2^ (Val)	SP	Method	Features	R^2^ (CV)	R^2^ (Val)
Ca	PLS	37,633	0.86	0.84	N	PLS	37,633	0.49	0.48
	PCA	242	0.87	0.85		PCA	242	0.58	0.56
	PCA	17	0.86	0.84		PCA	17	0.57	0.56
	CARS	1012/61	0.80	0.84		CARS	5/8	0.56	0.64
	Lasso	14	0.65			Lasso	36	0.44	
Mg	PLS	37,633	0.77	0.74	SOM	PLS	37,633	0.64	0.62
	PCA	242	0.80	0.78		PCA	242	0.65	0.65
	PCA	17	0.79	0.76		PCA	17	0.65	0.65
	CARS	22/61	0.90	0.80		CARS	12/7	0.64	0.69
	Lasso	9	0.53			Lasso	12	0.46	
K	PLS	37,633	0.92	0.91	Silt	PLS	37,633	0.88	0.86
	PCA	242	0.93	0.92		PCA	242	0.87	0.87
	PCA	17	0.92	0.91		PCA	17	0.87	0.85
	CARS	7/61	0.93	0.96		CARS	7/453	0.89	0.90
	Lasso	21	0.83			Lasso	13	0.73	
P	PLS	37,633	0.72	0.69	Clay	PLS	37,633	0.93	0.90
	PCA	242	0.72	0.68		PCA	242	0.93	0.91
	PCA	17	0.72	0.61		PCA	17	0.92	0.90
	CARS	6172/4130	0.75	0.75		CARS	7545/22	0.92	0.91
	Lasso	11	0.40			Lasso	8	0.81	

## Data Availability

Not applicable.

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
