# Peer review of "Mobile Laser-Induced Breakdown Spectroscopy for Future Application in Precision Agriculture—A Case Study"

_sensors, 2023, doi:10.3390/s23167178_

Round 1
Reviewer 1 Report
The authors reported the LIBS analysis of soil samples using several LIBS apparatuses and several kinds of analysis methods. The selection of apparatuses and analysis methods are adequate and discussed reasonably. Several points should be addressed before publication.
(1) In general, LIBS could not provide the contents of elements. However, Figures 1 and 2 indicate the correlation of contents obtained by LIBS. The method for evaluating the contents of element should be specified.
(2) From Table 2, it is difficult to compare the performance. The authors should specify how to evaluate the figures of merit from these data.
(3) Figure3 should be revised: the left figure could not be seen due to overlapping the right figure.
(4) HH would be the abbreviation of Handheld. It appeared at 179th line without specification (Page4), but spelled out form appeared at 183 line.
Reviewer 2 Report
General comments
The paper is about LIBS spectroscopy applied to soil samples.Portable and benchtop spectrometers were compared. The general idea is promising but some points must be clarified.
I understand the precision agriculture problematic and the tentative that are been conducted in order to use sensor in situ. Several sensors have been tested in different wavelength range (MIR, NIR, X-ray). I consider that LIBS is very promising. However, your study has a serious draw back that is the sample preparation. Which is the advantage of using portable LIBS if is required to mill the samples in a ball mill in order to prepare a pellet? How you suggest to apply it in the field? It must be highlighted in the introduction.
Another critical point in your study is the use of linear and non-linear regression methods. Why it was not evaluated the data set linearity? (See https://doi.org/10.1016/j.aca.2015.01.017 , https://doi.org/10.1016/j.sab.2021.106303) Moreover, you attest that the different regression models produced similar results, so probably non-linear regression methods are not required. You must review it.
The next critical point in data analysis is the spectrometers comparison. To compare the performance of the models by comparing the R2 (CV) and R2(Val) is a qualitative way. How you attest that a difference between 0.82 and 0.86 guarantees similar performance? A randomization test is a quantitative way to perform these comparisons. Both papers mentioned before brings details of the randomization test.
A very positive point in your manuscript is the models interpretability with VIP scores. Few researchers report it and it is crucial from the academic point of view in order to justify the models that will be used commercially in the near future.
Specific points
In the title the word “application” appears twice. Review it
Latin terms like, in situ, e.g, must be in italics
Page 3 line 109: One of the objectives is to emphasize “methodical extension”. What do you mean? In which parts of the manuscript it was considered? And in the conclusion?
Page 7 line 325: Have you considered Nitrogen? It appears for the first time in this sentence. How you considered it?
Page 9: Is there something wrong with figure 3. The red dots do not appear.
Conclusion: It must be completed rewritten according to the general comments, specially the terms: “good accuracy” (based in which statistical test?); “offers grate opportunities for precision agriculture” (but you need a ball mill and pressed pellets!), “regression methods lead to similar results” (but only qualitative comparisons were performed!).
The language needs revision. Specially the grammar in some sentences.
Round 2
Reviewer 2 Report
Dear authors,
Thank you for the replies.
I accept your arguments and consider that the paper is ready for publication.
I'm not native in English.
Minor revisions may be necessary.